

# Sexual dimorphism in adult Little Stints (*Calidris minuta*) revealed by DNA sexing and discriminant analysis

Aleksandra Niemc[1], Magdalena Remisiewicz[1,2], Joel Avni[3] and Les G. Underhill[2]

[1] Bird Migration Research Station, Faculty of Biology, University of Gdańsk, Gdańsk, Poland
[2] Animal Demography Unit, Department of Biological Sciences, University of Cape Town, Cape Town, South Africa
[3] Bird's-Eye View Productions, Kommetjie, South Africa

Corresponding author
Aleksandra Niemc,
aleksandra.niemc@phdstud.ug.edu.pl

## ABSTRACT

**Background**. The sex of an individual organism plays such an important role in its life cycle that researchers must know a bird's sex to interpret key aspects of its biology. The sexes of dimorphic species can be easily distinguished, but sexing monomorphic bird species often requires expensive and time-consuming molecular methods. The Little Stint (*Calidris minuta*) is a numerous species, monomorphic in plumage but showing a small degree of reversed sexual size dimorphism. Females are larger than males but the ranges of their measurements overlap, making Little Stints difficult to sex in the field. Our aim was to develop reliable sexing criteria for Little Stints in different stages of primary moult during their stay on the non-breeding grounds in South Africa using DNA-sexed individuals and discriminant function analysis.

**Methods**. We caught 348 adult Little Stints in 2008–2016 on their non-breeding grounds at Barberspan Bird Sanctuary. To molecularly identify the birds' sex we used P2/P8 primers and DNA isolated from blood samples collected in the field. We used Storer's dimorphism index to assess the degree of sexual size dimorphism. Then we divided our sample into two groups: before or during and after primary moult. For each group we developed two functions: one using wing length only and the other a combination of morphometric features including wing, tarsus and total head length. Then we used a stepwise procedure to check which combination of measurements best discriminated sexes. To validate our result we used a jack-knife cross-validation procedure and Cohen-kappa statistics.

**Results**. All the morphometric features we measured were bigger in DNA-sexed females than in males. Birds with fresh primaries had on average 2.3 mm longer wings than those with worn primaries. A discriminant function using wing length ($D_1$) correctly sexed 78.8% of individuals before moult, and a stepwise analysis showed that a combination of wing length and tarsus ($D_2$) correctly identified the sex of 82.7% of these birds. For birds with freshly moulted primaries a function using wing length ($D_3$) correctly classified 83.4% of the individuals, and a stepwise analysis revealed that wing and total head length ($D_4$) classified 84.7%.

**Discussion**. Sexual size differences in Little Stints might be linked to their phylogenetics and breeding biology. Females are bigger, which increases their fecundity; males are smaller, which increases their manoeuverability during display flights and hence their mating success. Little Stints show an extreme lack of breeding site fidelity so we did not expect a geographical cline in their biometrics. Sexing criteria available for Little

Stints in the literature were developed using museum specimens, which often shrink, leading to misclassification of live birds. The sexing criteria we developed can be used for studies on Little Stints at their non-breeding grounds and on past data, but should be applied cautiously because of the overlapping ranges.

# INTRODUCTION

An individual's sex is one of the most important factors shaping its biology. Male and female birds are subject to sex-specific selection pressures that entail differences in their biology, including migration strategies (*Remisiewicz & Wennerberg, 2006*; *Jakubas et al., 2014*), population structure (*Nebel, 2006*), foraging behaviour (*Mathot & Elner, 2004*; *Nebel, 2005*), moult (*Barshep et al., 2013*) and physiology (*Kulaszewicz, Wojczulanis-Jakubas & Jakubas, 2015*). Differences in the biology of males and females lead to diverging body sizes (*Fairbairn, 2007*) and are expected to emerge if selection for a character is stronger in one sex than in the other (*Székely, Lislevand & Figuerola, 2007*). Sexing monomorphic birds is difficult in the field, but can be done with molecular methods (*Dubiec & Zagalska-Neubauer, 2006*) using DNA isolated from blood samples (*Owen, 2011*), feathers (*Bello, Francino & Sánchez, 2001*) or buccal swabs (*Handel et al., 2006*); however, those methods often stress the birds and are expensive. For species in which males and females are monomorphic in plumage but show sexual size dimorphism (SSD) the sex can often be identified using morphometrics (*Dechaume-Moncharmont, Monceau & Cézilly, 2011*). In such cases the degree of sexual size dimorphism is crucial because in some species measurements overlap between sexes, which might lead to misidentification (*Jiménez, García-Lau & Gonzalez, 2015*). Discriminant function analysis enables observers to use a combination of morphological measurements to predict the sex of a bird with a certain probability (*Tabachnick & Fidell, 2001*). However, the efficiency of this method depends on the accuracy of the measurements, which suffers if researchers are not calibrated with each other or if wing length is measured on feathers in different stages of wear (*Dechaume-Moncharmont, Monceau & Cézilly, 2011*). Our study focuses on Little Stints, which are monomorphic in plumage but exhibit a small degree of sexual size dimorphism, with the females slightly larger than the males (*Del Hoyo, Elliott & Sargatal, 1996*). The ranges of their measurements overlap, which impedes sexing them in the field (*Prater, Marchant & Vuorinen, 1977*). Sexing criteria for Little Stints in the literature are usually based on small samples or on museum specimens, which are known to shrink (*Prater, Marchant & Vuorinen, 1977*).

We aimed to develop reliable sexing criteria for adult Little Stints by discriminant functions that accounted for the stage of wear of their primaries using measurements of DNA-sexed individuals spending the non-breeding season in South Africa. We suggest that

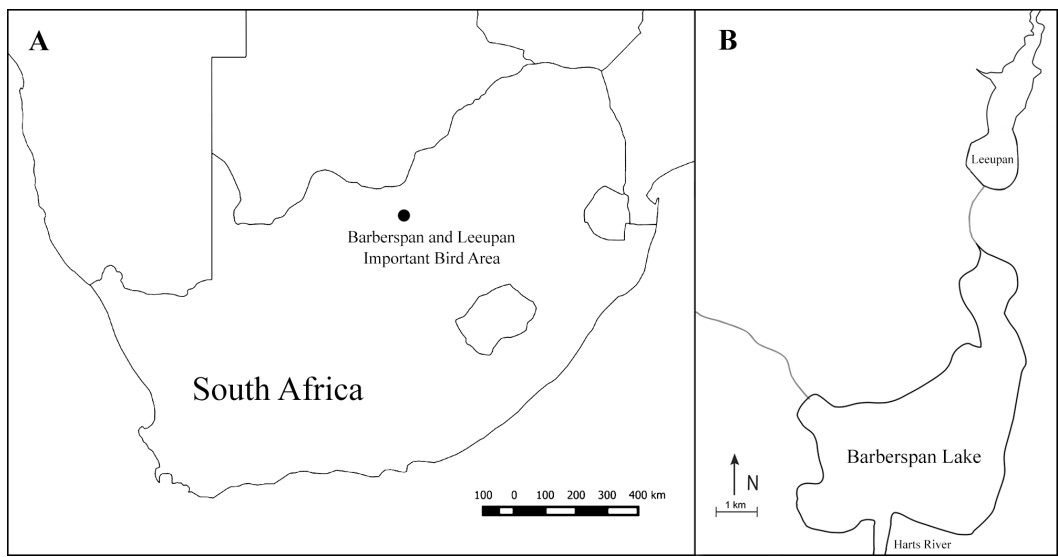

**Figure 1** **Map of study area.** (A) Location of Barberspan and Leeupan Important Bird Area. (B) Outline of Barberspan and Leeupan lakes. Figure credit: Jacek Mazur.

these discriminant functions serve as a useful tool, after adjustment to local measurement routines, for other researchers studying Little Stints at various stages of their life cycle.

## MATERIALS & METHODS

### Species and study area

The Little Stint is a long-distance migrant wader, which breeds in the Eurasian tundra and spends the non-breeding season from the Mediterranean coasts south throughout sub-Saharan Africa to South Africa in the south and southwestern Asia in the east (*Del Hoyo, Elliott & Sargatal, 1996*; *Tulp et al., 2002*; *Round et al., 2012*). A few individuals are also found further east along the East Asian–Australasian Flyway (*Tomkovich, Chih-Yuan & Liu, 2009*). Little Stint is a numerous species with an estimated world population of 1,500,000 individuals, a Least Concern conservation status and an increasing population trend (*Delany & Scott, 2006*; *Bird Life International, 2016*). Most Little Stints leave the non-breeding grounds and attempt breeding at the end of their first year, but a few stay in South Africa over the austral winter (*Underhill, 1997*; *Underhill et al., 1999*). Immature Little Stints can be distinguished from adults by the retained chestnut-fringed inner median coverts, which in adults are pale grey-brown tipped whitish (*Prater, Marchant & Vuorinen, 1977*). After arrival at the non-breeding grounds immature and adult birds undergo a complete moult, including the replacement of remiges. They complete this moult in January–March, after which the immatures become indistinguishable from the adults (*Prater, Marchant & Vuorinen, 1977*).

We caught Little Stints at Barberspan Bird Sanctuary (26°33′S, 25°36′E; North West Province, South Africa, Fig. 1). This reserve is centered on a shallow alkaline lake whose area varies from 257 ha to 2,000 ha, depending on rainfall (*Milstein, 1975*; *Barnes, 1998*). Lake

Barberspan is fed by the Harts River. In the austral winter and during droughts it becomes the only permanent waterbody in the surrounding farmland, which makes it an important stopover, moulting and non-breeding destination for waterbirds, including Palearctic migrants (*Barnes, 1998*; *Lipshutz et al., 2011*; *Remisiewicz & Avni, 2011*). Barberspan Bird Sanctuary is a Wetland of International Importance in terms of the Ramsar Convention and is an Important Bird Area according to Bird Life International (*Cowan, 1995*; *Barnes, 1998*; *Remisiewicz & Avni, 2011*).

## Data collection

During September–April in 2008–2016 we captured 348 adult Little Stints using walk-in traps (*Busse & Meissner, 2015*) and mist nets. We ringed and measured each bird. Measurements included: bill length (from the feather line to the bill tip), total head length (from the back of the skull to the bill tip) and tarsus length (from the tarsal joint to the distal end of the tarso-metatarsus), taken with callipers of 0.1 mm accuracy, and wing length (flattened and straightened wing, as in *Evans (1986)* and *Prater, Marchant & Vuorinen (1977)*), tarsus-plus-toe (*Anderson, 1975*) taken to the nearest 1 mm with a ruler, and body mass to 1 g or 0.1 g in different years (weighed with an electronic scale). We used only morphological measurements taken by MR (SAFRING ringing permit 1454), as recommended in *Henry et al. (2015)* because measurements taken by different ringers can reduce the accuracy of sex discrimination. We also took blood samples from a brachial vein (*Owen, 2011*) from all ringed Little Stints and preserved the samples in 96% ethanol for DNA sexing. Only trained, experienced team members handled the birds and took the blood samples to ensure safety standards. All the procedures were approved by the management of Barberspan Bird Sanctuary, under permits from SAFRING and the Department of Rural, Environmental and Agricultural Development, North West Provincial Government, South Africa.

## Molecular sex identification

We isolated DNA from blood samples using the Blood Mini Kit (A&A Biotechnology, Gdynia, Poland). The next step was PCR with the pair of P2 and P8 primers (*Griffiths et al., 1998*) using a modified protocol. Total volume of PCR was 20 $\mu$l, the reaction mix included: 7.5 $\mu$l REDTaq$^{\circledR}$ ReadyMix$^{TM}$ (Sigma Aldrich, St. Louis, MO, USA), 3.5 $\mu$l of water, 1 $\mu$l MgCl$_2$, 1 $\mu$l P2 primer (10 $\mu$M), 1 $\mu$l P8 primer (10 $\mu$M) and 2 $\mu$l of the DNA sample. For PCR amplifications we used an Eppendorf Mastercycler with this thermic profile: an initialisation at 94 °C for 2 min, 40 cycles of denaturation at 94 °C for 30 s, 40 cycles of annealing at 50 °C for 30 s and 40 cycles of elongation at 72 °C for 2 min, ending with a final elongation at 72 °C for 5 min. This method is based on the amplification of chromo-helicase-DNA-binding (CHD) genes found on avian sex chromosomes. The CHD-Z gene (ca 370 bp) is located on the Z chromosome, therefore it is present in both sexes. The CHD-W gene (ca 390 bp) occurs only on chromosome W, therefore it is unique to females (*Griffiths et al., 1998*). We separated the products of PCR with electrophoresis on 3.5% agarose gel (75 V, 120 min) stained with ethidium bromide (samples from 2008–2013) and Midori Green Advanced DNA Stain (NIPPON Genetics, samples from 2013–2016).

Products were visualised in UV light, one band was visible for males, which indicated ZZ chromosomes, and two bands for females (ZW chromosomes). The method enabled us to sex all birds sampled.

## Statistical analysis

For each morphometric measurement we used Storer's dimorphism index (SDI) to assess the degree of sexual size dimorphism in Little Stints (*Storer, 1966*), using the formula:

$$SDI = \frac{\text{mean} \female - \text{mean} \male}{(\text{mean} \male + \text{mean} \female) * 0.5} * 100.$$

We compared all the morphometrics we had measured of males and females using the two-sample $t$-test. Then we divided the birds into two groups: those before and those after primary moult, and compared the measurements of birds from these two groups. Birds caught in active moult were classified as "before moult" because they moult their outermost primary last. However, we did not measure wing length of any bird whose outermost primary was heavily damaged or if it was growing. For each group we used discriminant function analysis to determine the best set of measurements for sexing Little Stints with a two-fold approach. First, we used wing length alone as a discriminant factor. The second approach used a stepwise method including other measurements after conducting pairwise correlation of all the measurements. We used only one of a pair of correlated measurements at a time in the stepwise procedure to avoid multicollinearity. We did not include the body mass of Little Stints, because it changes during the non-breeding season during pre-migratory fuelling. The aim of producing two different discriminant functions for each group was to make these functions applicable for different data sets, because wing length is the most commonly taken measurement, in contrast to tarsus and total head lengths. We present the discriminant functions developed using different sets of morphometrics in the Supplemental Information. All the assumptions of discriminant function analysis were met (*Tabachnick & Fidell, 2001*), including the homogeneity of covariances (Box's M test), the homogeneity of variance (Levene's test), and the normal distributions of the measurements for males and females separately in each of the two groups. We confirmed no multicollinearity of the selected measurements ($r < 0.50$ for all pairwise correlations). We computed prior classification probabilities from the group sizes because of the unequal number of males and females in our sample (*Tabachnick & Fidell, 2001*). To validate our models we used a jack-knife procedure to assess the percentage of correctly sexed individuals by discriminant function analysis (*Dechaume-Moncharmont, Monceau & Cézilly, 2011*). This cross-validation technique predicts the sex of each individual using a discriminant function calculated for all the birds except the individual being classified (*Hair Jr et al., 1995*). We had unequal samples of males and females, so we assessed the effectiveness of our proposed functions by calculating Cohen's kappa statistic (*Titus & Mosher, 1984*), which estimates the improvement made by the results of discriminant analysis over random chance: 0 = no improvement over chance, 1 = full compliance (*Titus & Mosher, 1984*). The optimal cutting score was calculated as a weighted average of the group centroids (*Hair Jr et al., 1995*). Statistical analyses were

**Table 1  Morphological features of adult male and female Little Stints.**

| Measurement | Females | | | Males | | | $t$ | $p$ | SDI |
|---|---|---|---|---|---|---|---|---|---|
| | N | Mean (SD) | Range | N | Mean (SD) | Range | | | |
| Wing length (mm) | | | | | | | | | |
| before primary moult | 70 | 99.9 (± 2.2) | 95–104 | 86 | 96.5 (± 2.0) | 92–101 | 10.32 | <0.001 | 3.46 |
| after primary moult | 80 | 102.2 (± 1.9) | 98–106 | 77 | 98.4 (± 2.0) | 94–102 | 12.36 | <0.001 | 3.79 |
| Total head length (mm) | 163 | 39.30 (± 1.06) | 36.0–42.3 | 185 | 38.26 (± 0.98) | 35.8–41.1 | 9.62 | <0.001 | 2.69 |
| Bill length (mm) | 163 | 18.11 (± 0.92) | 16.3–20.5 | 185 | 17.32 (± 0.81) | 15.1–20.1 | 8.55 | <0.001 | 4.48 |
| Tarsus length (mm) | 162 | 22.09 (± 0.86) | 19.7–24.2 | 185 | 21.55 (± 0.78) | 19.4–24.4 | 6.16 | <0.001 | 2.47 |
| Tarsus-plus-toe length (mm) | 163 | 40.94 (± 1.27) | 37–44 | 185 | 40.15 (± 1.34) | 37–46 | 5.69 | <0.001 | 1.96 |
| Weight (g) | 162 | 24.48 (± 3.66) | 19–42 | 185 | 21.98 (± 2.43) | 17–33 | 7.42 | <0.001 | 10.78 |

Notes.

$t, p$, results of $t$-test comparing the sexes; SDI, Storer's dimorphism index.

performed in IBM SPSS Statistics for Windows, version 22.0 (IBM Corp., Armonk, N.Y., USA). All tests were two-tailed and the accepted level of significance was $P < 0.05$.

# RESULTS

## Morphological differences between the sexes

We identified 185 males and 163 females using DNA sexing. Analysis of morphometrics and Storer's dimorphism index (SDI) also revealed sexual differences. On average females were bigger than males in all morphological measurements (Table 1). The most dimorphic features were respectively bill, wing and total head lengths (Table 1). In all our measurements birds before and after wing moult differed only in wing length ($t_{311} = 7.69$, $P < 0.001$), which was on average 2.3 mm longer in those with fresh primaries after moult than in those before moult with worn primaries. We therefore conducted the discriminant analyses separately for these two groups.

Some morphological measurements were correlated (correlation coefficients for males and females between tarsus and tarsus-plus-toe were $r = 0.73$ and $r = 0.79$ and for culmen and total head length $r = 0.81$ and $r = 0.83$). We chose wing length, total head length (because it is less prone to errors than bill length measured to the feather line, which might be worn difficult to determine *Prater, Marchant & Vuorinen, 1977*), and tarsus length as the best factors for discriminant analysis.

## Discriminant functions for adult Little Stints before primary moult

Using measurements of 156 adult Little Stints (70 females and 86 males) taken before they had moulted their primaries, and using only wing length as a discriminant factor, we obtained the equation:

$$D_1 = -47.496 + 0.484 \text{ (wing)},$$

which allowed us to correctly classify 78.8% of the birds. A jack-knife cross-validation procedure yielded the same success rate, and our random chance-corrected procedure showed that our proposed classification was 56.8% better than chance (kappa = 0.568

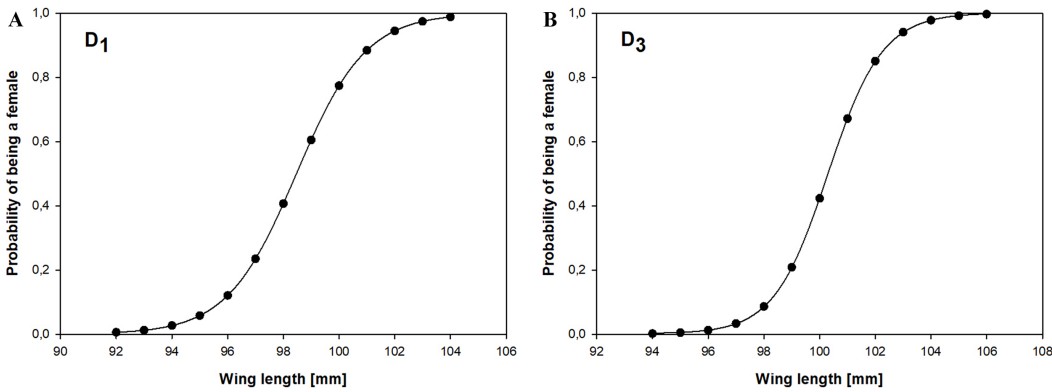

**Figure 2 Probability of being a female in relation to the wing length according to the discriminant function analysis for Little Stints.** (A) Birds before primary moult ($D_1$ function). (B) Birds after primary moult ($D_2$ function).

±0.082 SE, $P < 0.001$). If $D_1 > 0.17$, the bird was classified as a female, and if $D_1 < 0.17$ as a male (Fig. 2).

In the stepwise procedure including three selected measurements (wing, tarsus and total head length), only the combination of wing and tarsus length was a significant discriminant factor. The best discriminant function we obtained was

$$D_2 = -50.428 + 0.421 \, (\text{wing}) + 0.420 \, (\text{tarsus}).$$

This function correctly classified 82.7% of the birds and was more accurate than the previous equation. The cross-validation procedure correctly classified 80.8% of individuals and the classification was 65.1% better than chance (kappa = 0.651 ±0.081 SE, $P < 0.001$). If $D_2 > 0.18$ then the individual was a female and if $D_2 < 0.18$ it was a male (Fig. 3).

## Discriminant functions for adult Little Stints after primary moult

The second group we analysed comprised 159 adult Little Stints (82 females and 77 males) with all new primaries after their complete post-breeding moult. The function obtained using only wing length was

$$D_3 = -52.184 + 0.520 \, (\text{wing}).$$

This function correctly classified 83.4% of the individual birds, which was confirmed by the cross-validation procedure. The proposed classification was 66.9% better than chance (kappa = 0.669 ± 0.080 SE, $P < 0.001$). When $D_3 > -0.038$ the individual was classified as a female, when $D_3 < -0.038$ as a male (Fig. 2).

The stepwise analysis revealed that the best discriminating combination of measurements was wing and total head length (THL), according to the equation

$$D_4 = -59.310 + 0.445 \, (\text{wing}) + 0.377 \, (\text{THL}).$$

This function correctly classified 84.7% of the individuals in the sample, which was the highest proportion of all the equations we present. The cross-validation procedure showed

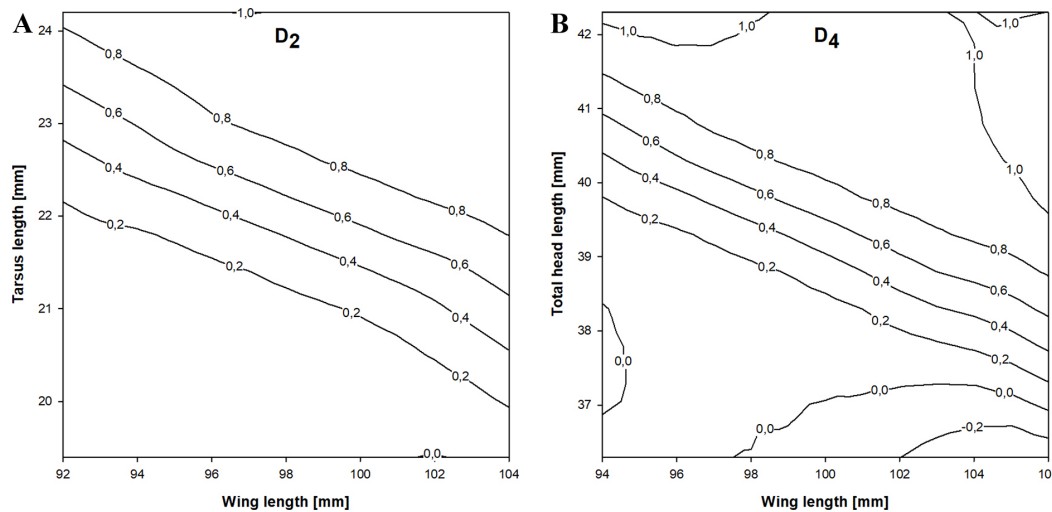

**Figure 3** **Probability of being a female in relation to the best combination of discriminating morpho-metric features for Little Stints.** (A) Birds before primary moult ($D_2$ function). (B) Birds after primary moult ($D_4$ function). Lines and values show the upper probability limits.

that the proposed equation was 84.1% correct and 69.4% better than chance (kappa = 0.694 ±0.080 SE, $P < 0.001$). If $D_4 > -0.041$ a Little Stint was classified as a female and if $D_4 < -0.041$ as a male (Fig. 3).

## DISCUSSION

We showed that the female Little Stints were larger than the males, not only in wing length, as described earlier (*Cramp & Simmons, 1983*; *Prater, Marchant & Vuorinen, 1977*), but also in other body features, as indicated by Storer's dimorphism index (Table 1). Thus, we recommend a combination of several measurements in discriminant functions as a useful tool in studies of sexual differences of this species. Such a size difference might benefit each sex in different ways, the females through increased fecundity and the males through higher mating success with smaller size, which we discuss.

Mean wing lengths based on the sexing criteria in *Prater, Marchant & Vuorinen (1977)* and established by *Tree (1974)* are 95.9 mm for adult males and 99.5 mm for adult females, 1.5 mm and 1.7 mm shorter than our results (Table 1). The sexing criteria presented in *Prater, Marchant & Vuorinen (1977)* were developed using museum specimens and thus subjected to shrinkage (*Jenni & Winkler, 1994*). We compared our differences with the one obtained using a regression equation in *Engelmoer et al. (1983)*, where: shrinkage = 0.006*fresh wing-length [mm] + 0.976, which predicted shrinkage for both sexes of 1.6 mm, similar to the difference we observed between the criteria in literature and for our sample. Moult and feather wear are important considerations when taking feathered measurements like wing length (*Meissner, 2005*; *Jiménez, García-Lau & Gonzalez, 2015*), thus we propose different discriminant functions for birds before and after moult. During the non-breeding season all Little Stints should be carefully examined for moult because
their outermost primary might still be growing. Measuring wing lengths of birds moulting P10 would underestimate wing length and could even cause misclassifications.

Sexual differences in the size of Little Stints might be closely related to their breeding biology. Male mating success is a strong form of sexual selection that affects their morphometrics. During the breeding season male Little Stints perform display flights, favouring smaller size to increase manoeuverability (*Figuerola, 1999*; *Székely, Reynolds & Figuerola, 2000*). The females' bigger size might be connected with increasing fecundity, because Little Stints exhibit successive bigamy of both sexes. Females lay two clutches in a short period (*Cramp & Simmons, 1983*; *Hildén, 1983*) and a larger body size allows them to compensate better for the increased energy expenditure of egg production (*Jönsson & Alerstam, 1990*). Sexual differences in morphometrics cannot be explained by a division in parental care, because each clutch and brood is cared for by a single parent of either sex (*Tulp et al., 2002*). The patterns we observed are in line with Rensch's rule, which points out that sexual differences in body size are usually small when females are the larger sex (*Rensch, 1950*; *Dale et al., 2007*).

Small calidridine sandpipers, such as Western Sandpiper (*Nebel, 2005*) and Least Sandpiper (*Nebel, 2006*), exhibit substantial sexual differences in bill length, which is the most dimorphic morphometric in Little Stints (Table 1). *Nebel & Thompson (2011)* show that sexual size dimorphism in calidrids is more distinct for trophic traits (e.g., bill length) than non-trophic traits, but those differences might be a result of their shared ancestry rather than natural or sexual selection.

A wide breeding range, such as that of the Little Stint, often results in a geographical variation in biometrics (*Zwarts et al., 1996*; *Dmitrenok et al., 2007*), which might distort a discriminant function analysis. We do not expect such differences in our study because Little Stints are an opportunistic species that show no natal philopatry or breeding-site fidelity and breed wherever they find favourable environmental conditions (*Hildén, 1983*; *Underhill et al., 1993*; *Tomkovich & Soloviev, 1994*). Their polygamous breeding system where males and females both frequently have two partners enhances gene-flow in the population, limiting geographical clines in the morphometrics of Little Stints.

## CONCLUSIONS

DNA sexing remains the most reliable method for monomorphic species, but discriminant functions are useful when researchers cannot collect or process DNA samples, get permits or when sex identification is needed in field studies. The functions we developed can be applied to measurements collected from Little Stints at their non-breeding grounds in the past and can strengthen the analysis when an individual bird's sex must be known. To make our functions more applicable for a wider range of researchers we present equations that can be used with different standard morphological measurements and equations that consider the wear of primary feathers. Our results are based on relatively large samples and all of the measurement were taken by one ringer, but ringing teams should regularly calibrate all people taking measurements (*Dechaume-Moncharmont, Monceau & Cézilly, 2011*), which improves the accuracy of sexing by discriminant functions. The functions

we suggest should be applied cautiously to data from other researchers because of the risk of misclassification, and should probably first be adjusted to each study, considering possible differences in measuring routines with those of our team. These functions should not be applied to Little Stints at their breeding grounds because their wing morphometrics possibly are different after migration. We suggest that the application of these or similar discriminant functions are a useful tool facilitating studies of differences in the biology of the sexes at different stages of the life cycle, in Little Stints and in other species exhibiting small sexual size dimorphism.

## ACKNOWLEDGEMENTS

We thank the staff at Barberspan Bird Sanctuary for their help and for making us feel welcome. We are grateful to all volunteers who helped to collect the data we used. We also thank Jacek Mazur for drawing the map and Pavel Tomkovich, Angelo Scherer and anonymus reviewer for their valuable comments on an earlier version of the manuscript.

### Funding

Our field expeditions were supported by the Polish Ministry of Science and Higher Education (SPUB grants) and Small Research Grants funded by the British Ornithologists Union. Data were analysed with funding from the University of Gdańsk and the Faculty of Biology and also grant PL-RPA/BEW/01/2016 funded by the National Centre for Research and Development (NCBiR) in Poland and the National Research Foundation (NRF) in South Africa, within the Poland-South Africa Agreement on Science and Technology. The funders had no role in study design, data collection and analysis, decision to publish, or preparation of the manuscript.

### Grant Disclosures

The following grant information was disclosed by the authors:
Polish Ministry of Science and Higher Education (SPUB grants).
British Ornithologists Union.
Faculty of Biology, University of Gdańsk.
National Centre for Research and Development (NCBiR) in Poland: PL-RPA/BEW/01/2016.
The National Research Foundation (NRF) in South Africa, within the Poland-South Africa Agreement on Science and Technology.

### Competing Interests

The authors declare there are no competing interests.

### Author Contributions

- Aleksandra Niemc performed the experiments, analyzed the data, contributed reagents/materials/analysis tools, prepared figures and/or tables, authored or reviewed drafts of the paper, approved the final draft.

- Magdalena Remisiewicz conceived and designed the experiments, performed the experiments, contributed reagents/materials/analysis tools, authored or reviewed drafts of the paper, approved the final draft.
- Joel Avni performed the experiments, authored or reviewed drafts of the paper, approved the final draft.
- Les G. Underhill contributed reagents/materials/analysis tools, authored or reviewed drafts of the paper, approved the final draft.

## Animal Ethics

The following information was supplied relating to ethical approvals (i.e., approving body and any reference numbers):

The Department of Agriculture, Conservation, Enviroment & Rural Development, North West Provincial Government, Republic of South Africa approved our research.

## Field Study Permissions

The following information was supplied relating to field study approvals (i.e., approving body and any reference numbers):

The approval of field experiments and ringing permits were provided by the South African Bird Ringing Unit (SAFRING) and Department of Agriculture, Conservation, Enviroment & Rural Development, North West Provincial Government, Republic of South Africa.

## Data Availability

The raw data are provided in a Supplemental File.

## Supplemental Information

Supplemental information for this article can be found online at http://dx.doi.org/10.7717/peerj.5367#supplemental-information.

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
