# Peer review of "Sexual dimorphism in adult Little Stints (Calidris minuta) revealed by DNA sexing and discriminant analysis"

_PeerJ, doi:10.7717/peerj.5367_

## Round 0.1 · original submission · Minor Revisions

The reviewers were all positive about your manuscript and it will be acceptable for publication after the relatively minor changes that they have identified, have been implemented. This will make a nice contribution to the literature in this area.

·

Basic reporting

no comment

Experimental design

no comment

Validity of the findings

no comment

Additional comments

This paper is clear advancement in sexing of an abundant and widely distributed wader species, the Little Stint, which is inadequately studied in many respects. The developed discriminant functions for sexing will be of great help for further studies of this species. The paper is clearly and well written. It does not evolve any criticism. But I have several small comments/suggestions.
Line 82: It is worth adding that it is not only long-distance species, but also abundant one.
Lines 83-84: The described non-breeding distribution reflects areas where most Little Stints are wintering, but in small numbers they can be found farther east (similarly with low densities in the east of their breeding range). See: Round P.D. et al. (Stilt, 2012, V61, Pp55-56) and Tomkovich et al. (Wader Study Group Bulletin, 2009, V116, No3, Pp199-201).
Lines 229-230: There is no need to repeat here that females have longer wings. This is said in the previous paragraph.
Lines 245-250: This speculation about reasons of the revealed sexual size dimorphism does not sound convincing to me, although I have nothing to suggest instead (this issue needs study). Little Stint males differ from most other sandpipers by reduced breeding territoriality and related displays. Females of this species are bigamous indeed, but they are bigger than males similarly with monogamous Calidris sandpipers. This means that probably not bigamy, but something else instead is a reason for females to be bigger.

Reviewer 2 ·

Basic reporting

.

Experimental design

.

Validity of the findings

.

Additional comments

This paper describes a solid piece of work, which should be published after revision. I have the following opinions and comments, which will need some thought, and further analyses if my comments are acceptable.
1. Abstract line 35. Change “plumage” to “primaries”.
2. There is both English and American spelling. See lines 52 “behavior” and 53 “moult”.
3. Line 86. “gap year” is slang. Remove.
4. Why were first-year birds ignored? Seems a shame just to do adults. Either explain why first-years were ignored (small sample), or include basic morphometrics to show that they are the same as adults, or carry out discriminant functions for them too.
5. Line 109. “chord” is wrong. This is the distance between the leading and trailing edge of a wing.
6. Line 109. The tarsus + toe measure is first described by Anderson (c. 1975) in Wildfowl on moorhens.
7. Line 142. Give months covering “before and after primary moult”. Further, if wing length decreases during the inter-moult period, this will affect both the discriminant functions, and their application. You need to show how and when wing length reduction occurs. For some species (red knot), it is a steady reduction – this would mean that each month needs to be adjusted to a given standard month. However, if there is no measurable wear between arrival and the start of mouth, and between the end of moult and northward migration, then there is no need for adjustment. Therefore, wing length change (accounting for sex) needs to be shown for the before and after moult periods. Note too, that application of your functions cannot apply to breeding birds. You need to say this.
8. Total head length is not as frequently measured as bill length. Therefore, your functions are unlikely to be used by others. Further, I would disagree that bill length has more error than total head length. Can you show that you are correct?
9. Likewise, tarsus length is a poorly defined parameter by most (between observer variation is large). Tarsus + toe is less problematic in this respect.
10. Following on from 8 and 9, you could provide functions that are more likely to be used by others by using the parameters that are most likely to be used by others, even although the power of discrimination is poorer (I would guess this would be minor). Otherwise, your study becomes one of limited application.
11. Line 192. What does D actually mean? What does it say for an individual. Ideally, one want’s to know the probability of a bird being male or female. One can then make a judgement on the value of that probability. I would accept a 0.9, but not 0.6.
12. Line 248. I don’t think it is called “successive bigamy”.
13. There is not consistency in the references. See line 372.
14. Line 254. The statement about which parameter is most dimorphic is incorrect – see table 1 - (4.48).
15. Table 1 needs to show wing length before and after moult.
16. Figures. What are the boxes, vertical lines? What do the +ve and –ve scores mean? I would want to see the 95% ranges (not CLs) for both species.

·

Basic reporting

In general the article is well written, concise and easy to read. Is in accordance with the norms of scientific writing.

The literature references were appropriately chosen for the theme. They can be complemented as indicated throughout the manuscript.
In the background, the authors introduce the text with the sexual selection (line 15, pg 1) which leads to understand in the context that is the one responsible for the RSSD observed in Little Stint (Calidris minuta). Additionally, at least the three largest measures of sexual dimorphism should be indicated in the abstract.
The authors should make it clear that the results should be used carefully because of the misclassification of sex (21.2 to 15.3%), according to the used FDA.

The structure of the article is in conform the suggested format and raw data are available.
Table 1 shows a mistake. Storer’s index is not calculated according de formula of the Methods section (all measurements).
Example: Wing SI on table 1 = 3,80
Wing = (101,19 – 97,42)/ (101,19 + 97,42) *100 Wing = 1,89
DIstorer^' s= (mean ♀-mean ♂)/(mean ♂+mean ♀)*100
Figures 1 and 2 are not required
Include a figure in Species and Study area section (pg 4, line 100) – The described area would be better represented on a map of the country with the study area.

The manuscript presents clear hypotheses and has been satisfactorily addressed in an objective and concise manner.

Experimental design

The study is original, examines an alternative of identifying the sex of monomorphic shorebird and fills a gap in Calidris minuta sexing that has not yet been studied.
I emphasize that the authors performed a great fieldwork with a large number of captured shorebirds, and a fine lab work with a great number of samples, which gives good credibility to the results.

The study was well conducted with a good sample size, careful collection of birds morphometric measurements, advanced techniques of DNA extraction and amplification of the gene of interest for the birds sexing.
International standards for capture, handling, sample collection and laboratory work were observed.

The methodology has been described in details and can be replicated easily elsewhere.
Pg 5, line 100 - It would be better represented on a map of the country with a study area;
Pg 5, line 115 – “and body mass to 1 g or 0.1 g in different years (weighed with an electronic scale)” – the body mass weighed to 1g has a low accuracy in measurement

Validity of the findings

The study is original and employs modern techniques for sex determination of birds, in the present case in Calidris minuta. It has application in every wintering area of the species in West Africa. In this sense, it has an important impact for other studies to be carried out with the species, as well as significantly reducing the effort and costs to carry out fieldwork that intend to determine the sex of the bird to identify specific patterns in these birds.

The large number of samples is a strong point of the manuscript, since it allowed the performance of statistical analyzes with high reliability and robustness. The sample collections and analyzes were obtained with care and standard procedures of accomplishment and control.

The conclusions are clearly presented and are in line with the proposed objectives. They also present the limitations of the presented results, which contributes to avoid future errors in the use and application of the same in other field studies. They could address and discuss a little more about the Storer’s Index that indicates RSSD in the species under study.

Unidentified speculation in the text

Additional comments

Please see further review comments throughout the manuscript in word document.
I congratulate the authors for the great effort in carrying out this work. It was eight years of catches of the species (probably many more species) with mist nets and walk-in traps, sample collection, DNA extraction, gene amplification and analyzes. All this carried out with high quality and use of procedures standardized and used internationally. Congratulations.

---

## Round 0.2 · accepted · Accept

You have done a very thorough job of revising the manuscript in line with the comments of the reviewers and the improved manuscript is now acceptable for publication. Well done on your nice study.

#